# Neuromorphic photonic computing with an electro-optic analog memory

Sean Lam [1] ✉, Ahmed Khaled [2], Simon Bilodeau[3], Bicky A. Marquez[2], Paul R. Prucnal [3], Lukas Chrostowski [1], Bhavin J. Shastri [2,3,4] ✉ & Sudip Shekhar [1] ✉

In neuromorphic photonic systems, device operations are typically governed by analog signals, necessitating digital-to-analog converters (DAC) and analog-to-digital converters (ADC). However, data movement between memory and these converters in conventional von Neumann architectures incur significant energy costs. We propose an analog electronic memory co-located with photonic computing units to eliminate repeated long-distance data movement. Here, we demonstrate a monolithically integrated neuromorphic photonic circuit with on-chip capacitive analog memory and evaluate its performance in machine learning for in situ training and inference using the MNIST dataset. Our analysis shows that integrating analog memory into a neuromorphic photonic architecture can achieve over 26 × power savings compared to conventional SRAM-DAC architectures. Furthermore, maintaining a minimum analog memory retention-to-network-latency ratio of 100 maintains >90% inference accuracy, enabling leaky analog memories without substantial performance degradation. This approach reduces reliance on DACs, minimizes data movement, and offers a scalable pathway toward energy-efficient, high-speed neuromorphic photonic computing.

## Motivation

Artificial intelligence (AI) has profoundly influenced society and technology with advancements ranging from ChatGPT's human-like dialogs (https://chat.openai.com/) to AlphaFold's protein structure predictions[1–3]. Historically, AI's progression has paralleled advances in CMOS electronics, driven by Moore's law and Dennard scaling[4]. While Moore's law has slowed down, AI's performance demands have surged, with computational needs doubling every 2 months[4]. In parallel, neuromorphic engineering has emerged as a field aiming to align AI algorithms to hardware architectures that mirror their inherently distributed nature, with demonstrated applications in particle physics, nonlinear programming, and signal processing[5]. Among these innovations, neuromorphic photonics promise high bandwidth and low latency, providing a complementary opportunity to extend the domain of AI[5–8]. The key advantages of neuromorphic photonic accelerators over purely electronic implementations include 1) massive parallelism using optical techniques, such as wavelength-division multiplexing (WDM), mode multiplexing (MDM), and time-domain multiplexing to carry multiple signals in a single interconnect and 2) lower energy-delay products in optical interconnects for multiply-accumulate (MAC) operations with potential speeds on the order of peta MACs per second per mm$^2$ and energy efficiencies in the atto joule per MAC regime[5,9–12]. However, most photonic processors have been limited to stationary weights, which are trained offline on digital computers and remain unchanged during inference, thus restricting their application potential. One reason this limitation arises is because analog photonic processors require analog-to-digital (ADC) and digital-to-analog (DAC) converters (Fig. 1b), which become critical

[1]Electrical and Computer Engineering, University of British Columbia, Vancouver, BC, Canada. [2]Centre for Nanophotonics, Physics, Engineering Physics & Astronomy, Queen's University, Kingston, ON, Canada. [3]Electrical and Computer Engineering, Princeton University, Princeton, NJ, USA. [4]Smith Engineering, Electrical and Computer Engineering, Queen's University, Kingston, ON, Canada. ✉e-mail: seanlm@student.ubc.ca; shastri@ieee.org; sudip@ece.ubc.ca

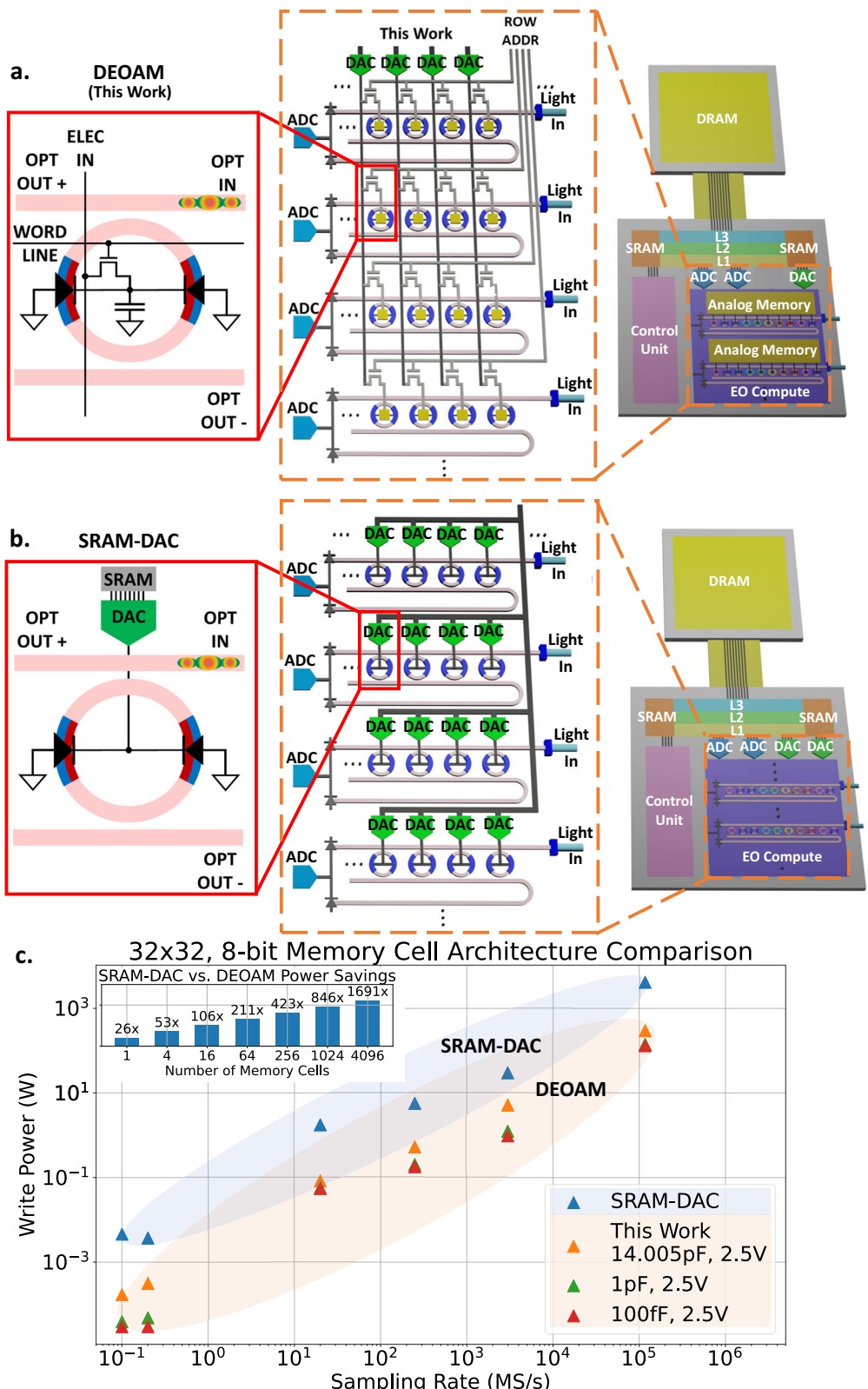

**Fig. 1 | Motivation for electro-optic processors with dynamic electro-optic analog memory (DEOAM). a** Dynamic electro-optic analog memory (DEOAM) consists of a capacitor connected to a PN junction microring resonator (MRR). The capacitor holds data on the MRR, thereby enacting a weight in the optical domain. Using DEOAM means DACs can be shared amongst columns of analog memory devices, updating them row by row since the analog memory holds the signal on the MRR, which relaxes limitations on energy efficiency and bandwidth imposed by DACs. **b** Conventional electro-optic processors require a dedicated SRAM-DAC for each PN junction MRR. The DAC needs to be constantly active to hold a signal on the MRR; therefore, implementing SRAM-DAC in a neuromorphic photonic processor requires a DAC for every MRR. **c** If $n$ represents the number of MRRs in the row and column of a neuromorphic photonic processor, scaling the conventional SRAM-DAC implementation incurs an $n^2$ DAC penalty, whereas DEOAM scales by $n$. Write power and sampling rate for $32 \times 32$, 8-bit memory cell architectures are compared using each memory device and DACs from literature[22–30]. As DEOAM scales, DEOAM saves more power compared to SRAM-DAC.

bottlenecks that affect bandwidth and energy efficiency as data transitions between digital memory and analog computation[5,13–15]. During training, backpropagation requires computations (gradient and error calculations) and memory accesses (weight updates and temporary data storage for calculations), meaning that any part of the pipeline that requires data converters to compute or access memory can be a bottleneck. This work focuses on data conversion bottlenecks from memory and showcases energy savings and minimized latency from co-locating analog memory with analog optical computational units.

Analog memory can reduce the number of DACs in neural network architectures (Fig. 1a). While optical analog memories (OAM) have been explored as in-memory computing devices, they currently face challenges, such as low yield and endurance, incompatibility with standard foundry processes, and complex fabrication processes[16,17]. Phase change materials (PCM) have been demonstrated in a photonic tensor core enabling $10^{12}$ MACs per second for inference tasks[9]. However, the endurance of PCMs can vary depending on the modulation mechanism (electrical, electrothermal, or optical), raising concerns about their suitability for neuromorphic training applications that typically require volatile memory[18]. Optical memristors, which store memory by forming and dissolving filaments, offer minimal footprint, making them promising candidates for large scale integration. However, their limited endurance (~2000 write cycles) and challenges in compatibility with standard foundry processes remain key obstacles to broader adoption[16,19]. In contrast, analog electronic memories, such as dynamic random-access memory (DRAM), metal-oxide-metal capacitors, metal-insulator-metal capacitors, or metal-on-semiconductor capacitors, are foundry-compatible and included in conventional CMOS technologies. This compatibility makes integrating analog electronic memories into neuromorphic photonic hardware a more viable approach, potentially reducing the number of ADCs and DACs in applications, such as long-short term memory neural networks[20].

Here, we propose a neuromorphic photonic processor with dynamic electro-optic analog memory (DEOAM, Fig. 1a). This processor leverages the monolithic integration of nanophotonic and CMOS devices on the same silicon substrate. The proposed architecture (Fig. 1a) merges the parallel information processing capabilities of photonics using wavelength-division multiplexing (WDM) with the low control complexity of crossbar arrays offered by electronics. As a proof-of-concept, we demonstrate neurosynaptic photonic weights, realized through a WDM array of tunable PN junction microring resonators (MRR) within a photonic weightbank and integrated with an array of capacitive DEOAM on the 90 nm GF9WG monolithic process. The MNIST dataset serves as a benchmark for training and inference and reveals tradeoffs in neural network design[21]. Our approach to integrating analog memory in neuromorphic photonic processors aims to create a flexible processor with high bandwidth, efficient training capabilities, and the ability to adapt through on-chip, online learning.

## Dynamic Electro-Optic Analog Memory (DEOAM)

DEOAM consists of a capacitor that holds charge on a reverse-biased PN junction MRR (Fig. 1a). Data is stored on the capacitor as an electrical signal then the PN junction MRR converts the signal to the optical domain for processing. DEOAM can be integrated into an array of MRRs, functioning as an electronic crossbar array and photonic weight banks (Fig. 1a) in a neuromorphic photonic processor. The conventional approach consists of several bits of static random access memory (SRAM) connected to a DAC that continuously drives a PN junction MRR (Fig. 1b).

In a neuromorphic photonic processor with $n$ rows and $n$ columns of MRRs acting as weights, DEOAM reduces the number of DACs, thereby reducing power consumption. Since the conventional SRAM-DAC implementation requires each MRR to be paired with a DAC,

scaling the implementation results in the number of DACs increasing by $n^2$ (Fig. 1b). The power consumed by SRAM-DAC is calculated as:

$$P_{\text{SRAM}-\text{DAC}} = n^2(P_{\text{DAC}_{\text{static}}} + P_{\text{SRAM}_{\text{static}}}) + n(P_{\text{DAC}_{\text{dynamic}}} + P_{\text{SRAM}_{\text{dynamic}}}) \quad (1)$$

where $P_{\text{SRAM}-\text{DAC}}$ is the total SRAM-DAC array power consumption in W, and $P_{\text{SRAM}}$ and $P_{\text{DAC}}$ are the SRAM and DAC power consumption in W, respectively, which includes static and dynamic power consumption and accounts for varying bit precisions of the DAC. All surveyed DACs are demonstrated, and the SRAMs associated with each DAC are surveyed to be in the same process node[22–30].

In contrast, using DEOAM reduces the number of DACs below $n^2$ (Fig. 1a) because data is held directly on the photonic device via analog memory, eliminating the constant processing typically needed by DACs. This concept is similar to the functionality in modern liquid crystal active-matrix displays[31]. The power consumed by DEOAM is calculated as:

$$P_{\text{DEOAM}} = n(P_{\text{DAC}_{\text{static}}} + P_{\text{DAC}_{\text{dynamic}}} + \alpha C f V^2) \quad (2)$$

where $P_{\text{DEOAM}}$ is the total DEOAM array power consumption, $\alpha$ is the activity factor (assumed to be 1), $C$ is the load capacitance in F, $f$ is the switching frequency in Hz and is half the sampling rate, and $V$ is the switching voltage. If we consider a memory element and DAC in both architectures, both architectures consume similar power when modulating the inputs (input to the SRAMs in the SRAM-DAC architecture and input to the DAC in the DEOAM architecture), but the main difference in power consumption is the SRAM to DAC power consumption in the SRAM-DAC architecture, the DAC to DEOAM power consumption in the DEOAM architecture, and the distribution of active DACs between the two architectures. In the SRAM-DAC architecture, power consumption begins when the SRAM changes state and ends when the DAC settles, meaning there is SRAM static and dynamic power and DAC static and dynamic power. In the DEOAM architecture, power consumption begins when the DAC changes state and ends when charge settles on the DEOAM, meaning there is DAC static and dynamic power and DEOAM dynamic power. Finally, there is the distribution of DACs from the architecture that determines the power consumption scaling. DACs are still required in the DEOAM architecture to load initial weights from non-volatile, digital memory before beginning inference or training and to refresh analog memory during inference. However, this approach not only simplifies control complexities by reducing the number of electrical lines needed for DAC operations but also, with longer retention times, diminishes the need for DACs to continuously drive the photonic devices, leading to further power savings as the processor scales (Fig. 1c, inset). As the DEOAM capacitance decreases, further power savings can be realized up to the limit of the junction capacitance of the PN junction (<50 fF[32,33]). Hence, integrating DEOAM and reducing the reliance on DACs paves the way for more efficient computing.

In neuromorphic processors, analog memory is typically situated close to or directly integrated with the computing unit to minimize data movement, thereby reducing energy consumption and latency[34,35]. Data movement in memory accesses can consume more than 10 times the energy of multiplication and addition operations[36]. Recent demonstrations of in-memory compute, such as Mythic's Analog Matrix Processor or NeuRRAM, achieve up to tens of TOPS/W in compute efficiency by co-locating the analog memory with the compute unit[9,37–39]. This highlights DEOAM's close integration with the MRR, which acts as the compute unit in the optical domain, thereby minimizing data movement and enhancing compute efficiency.

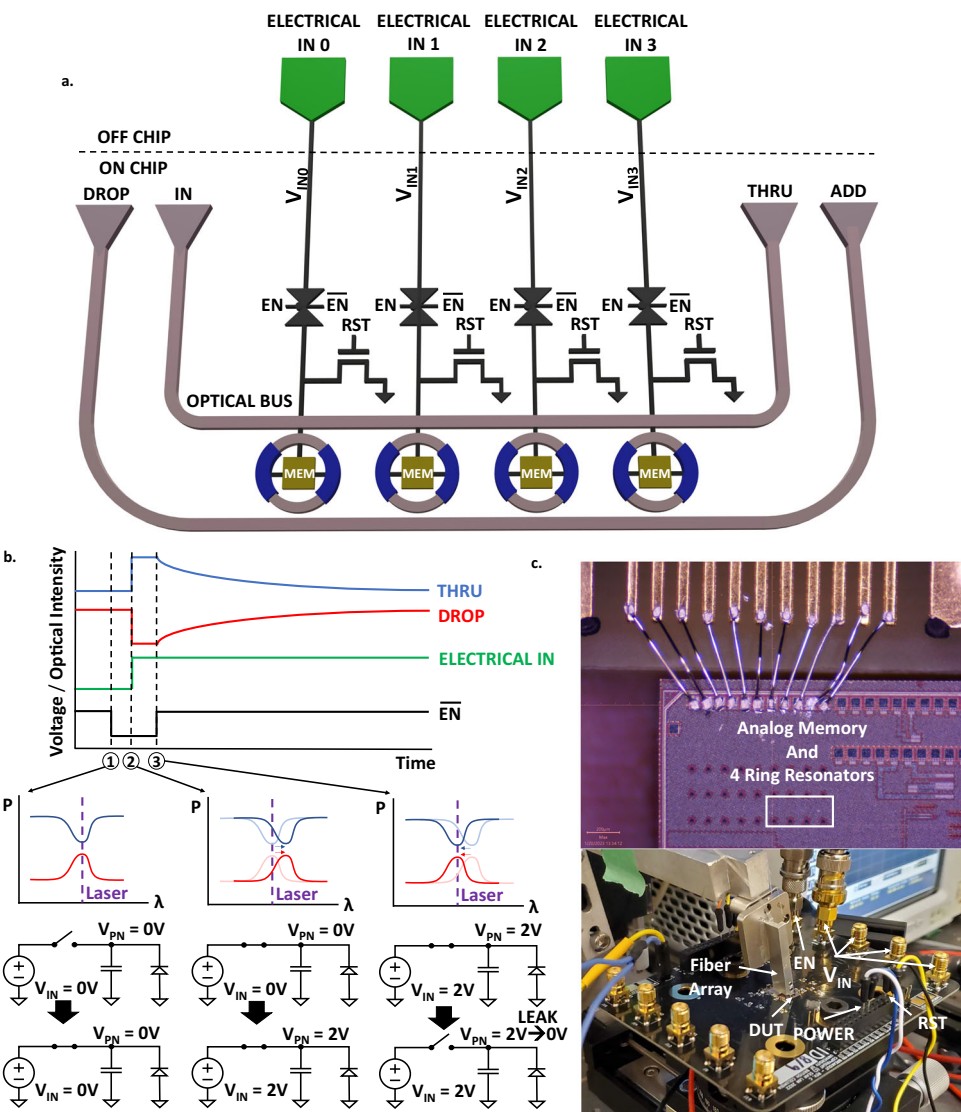

**Fig. 2 | The architecture and operation of the neuromorphic photonic proto-type with integrated DEOAM. a** The analog memory circuit consists of a 1x4 MRR weight bank with each MRR connected to a capacitive analog memory. Transmission gates are used to enable/disable (EN) writing and switches are used to reset (RST) the analog memories. Laser light is injected into grating couplers through the IN port and is measured at the THRU and DROP ports. **b** The operation of the DEOAM circuit is shown. The PN junction MRR is modeled as a reversed biased diode, and the transmission gate is modeled as an ideal switch. At time 1, the laser has been tuned to the MRR's resonance, and the transmission gate is connected such that the voltage source ($V_{IN}$) can drive the analog memory ($V_{PN}$). At time 2, voltage is driven onto the capacitor and MRR, causing the MRR's resonance to shift. At time 3, the transmission gate disconnects the voltage source from the capacitor, and the capacitor holds the charge on the MRR until it leaks away, returning the MRR resonance back to its original state. **c** The chip is wire bonded to a custom printed circuit board with thermal control and high-speed electrical lines. Details about the hardware operation and measurement setup are expanded in "Methods".

## Results

### Electro-optic characteristics

The neuromorphic photonic hardware with integrated DEOAM is shown in Fig. 2. The MRRs are PN-junction-doped modulators operating in depletion mode (reverse bias) and are connected to analog memory cells consisting of $100 \times 100\,\mu m$ metal-oxide-metal capacitors that use interdigitated fingers to increase capacitance. The MRRs have different ring perimeters starting with $144.248\,\mu m$ and $50\,\mu m$ PN junctions, increasing by 50 nm to avoid resonance collision. Key figures of merit for DEOAM are summarized in Table 1, and measurement results are shown in Fig. 3. Figure 2a, b show the DEOAM circuit and its operation. Weights are loaded as voltages on the capacitive memory, affecting the MRR's resonance. The capacitive memory holds the voltage on the MRR until the charge leaks away, returning the MRR to its resonance. Figure 2c shows the chip and measurement setup for static

optical and time domain measurements. Free spectral range, inter-channel spacing, Q-factor, and extinction ratio are crucial optical characteristics that determine the maximum number of MRRs and weights in a weight bank in a specific wavelength range. Therefore, if additional weights are needed in each row of a weight bank, solutions like adding more cores, reusing hardware, or employing time multi-plexing may be needed for processing neural network data. In a neu-romorphic photonic weight bank, light modulation is initiated at the input data modulators. The light then travels through the weight banks before being absorbed by photodetectors (PD) and amplified by transimpedance amplifiers (TIA). The group index helps characterize the group delay, which determines the overall compute time when combined with the delay from the PD and TIA. Since the delay from the PD and TIA depends on their design and technology, we report com-pute time based primarily on the group delay. State-of-the-art

**Table 1 | Experimental measurement results of the electro-optic analog memory circuit**

| Parameter | Symbol | Min. Value | Nom. Value | Max. Value | Unit |
|---|---|---|---|---|---|
| Free Spectral Range | $FSR$ | - | 4.67 | - | nm |
| Full Width Half Max. | $FWHM$ | - | 0.019 | - | nm |
| Finesse | $F$ | - | 245.79 | - | - |
| Inter-channel Spacing | $\Delta\lambda_O$ | 0.34 | - | 2.61 | nm |
| Group Index | $n_g$ | 3.70 | 3.70 | 3.71 | - |
| Q-Factor | $Q$ | 46050 | 49234 | 51484 | - |
| Extinction Ratio | $ER$ | 15.6 | - | 19.9 | dB |
| Wavelength Tuning Efficiency | - | - | 6.22 | - | pm/V |
| Modulation Efficiency | - | - | 0.23 | | dB/V |
| Retention Time (10–90% Level) | $t_{ret}$ | 0.3330 | 0.5735 | 0.7957 | ms |
| Retention Time (One Time Constant) | $\tau_{ret}$ | 0.2590 | 0.5527 | 0.8345 | ms |
| Compute Time[a] | $\tau_{read}$ | 70.87 | - | 79.51 | ps |
| Write Time - Rising Edge (10-90% Level) | $t_{rise}$ | 65.5 | 75.3 | 91.3 | ns |
| Write Time - Falling Edge (10-90% Level) | $t_{fall}$ | 76.2 | 89.4 | 121.2 | ns |
| Write Time - Rising Edge (One Time Constant) | $\tau_{rise}$ | 35.5 | 43.8 | 52.0 | ns |
| Write Time - Falling Edge (One Time Constant) | $\tau_{fall}$ | 51.0 | 63.3 | 71.7 | ns |
| Write Energy Consumption[b] | $\eta$ | 26.82 | 55.97 | 80.97 | pJ/Write |
| Bit Precision | $N_b$ | 5.33 | 5.65 | 5.97 | bits |
| Capacitance | $C_{MEM}$ | - | 14.005 | - | pF |

Write time and retention time measured with 0.5 dBm optical bus power in the circuit. Details regarding measurement setup and optical characteristics can be found in Methods and Supplementary Information.

[a] Estimated by the group delay through the circuit and resonance build-up in the MRRs. The summation of path delay ($\tau_{path} = Ln_g/c$ where $\tau_{path}$ is the path delay, $L$ is the path length, and $c$ is the speed of light) and MRR resonance build-up time ($\tau_{MRR} = FRn_g/c$ where $R$ is the ring radius) defines compute time[73].

[b] Write energy consumption derived from maximum, minimum, and nominal retention time curves with nominal leakage curve at 0.5 dBm optical bus power in the circuit.

demonstrations of PDs and TIAs yield latencies ranging in tens of ps[40–44]. The wavelength and modulation efficiencies are also important for characterizing the tunability and range of weights for the neural network.

Key FoMs to characterize the capacitive analog memory are retention time, write time, read time, and power consumption. Since data written to DEOAM is immediately available in the optical domain, read time is effectively zero. Retention time is dependent on leakage, and leakage is dependent on optical power and photoresponse of the PN junction MRR[45]. Figure 3a, b reveals that the retention time of 0.8345 ms for one time constant can be extended if lower optical power is used since lower leakage is observed, but this involves a tradeoff between retention time and signal-to-noise ratio. Energy consumption, estimated to be 55.97 pJ/write, is linked to capacitor leakage and charge replenishment to maintain weight values. Figure 3c, d, and e shows that the write time is about 40−50 ns for one time constant, enabling MHz write speeds. From Fig. 3, the analog memory bit precision is about 5 bits and can be extracted from the time domain responses based on $\log_2(\frac{\overline{\mu}_{max} - \overline{\mu}_{min}}{\sigma}) = N_b$ where $\overline{\mu}_{max}$ and $\overline{\mu}_{min}$ are the mean values at the maximum and minimum range of the analog memory, respectively, $\sigma$ is the standard deviation, and $N_b$ is the bit precision of the analog memory[46].

## Analysis of analog memory specifications

To verify different analog memory specifications within a neural network system, a weight bank architecture, depicted in Fig. 4, is emulated to assess analog memory performance during inference and training using the MNIST dataset[21]. The neural network architecture is configured as a three-layer model. The input layer processes 784 values corresponding to 28 × 28 pixel image. This is followed by a hidden layer with 50 neurons and a ReLU activation function, and finally, an output layer with 10 neurons and a logarithmic softmax activation for digit classification. The network achieves over 95% validation and testing accuracy across all predicted digits within one

training epoch, as shown in Fig. 4b, which is comparable to the photonic neural network demonstrated in ref. 47. For a 5-bit resolution MRR-based weight bank matrix incorporating SOAs and MRRs with a finesse of 368 (≈55,000 quality factor), the network size is limited to 108 rings (wavelength channels) per weight bank due to the MRRs' FSR and finesse. The number of rows (spatial channels) is limited to 60 due to the signal-to-noise ratio (SNR) constraints[48,49]. To ensure compatibility within a single core, the implementation uses 80 MRRs and 80 analog memories arranged in 50 rows of weight banks. Each weight bank requires two PDs and one TIA to perform the summation and signal amplification.

In Fig. 5, the weight bank array is characterized for inference accuracy by comparing inference accuracy using weights trained on a GPU with full floating point accuracy and weights trained on emulated hardware with non-idealities in the analog memory and hardware. These non-idealities include analog memory control bit precision, noise, latency, and retention time, which can be generalized to any analog memory technology. These non-idealities are examined individually to assess their impact on inference accuracy. Figure 5a reveals that to achieve inference accuracies above 80%, more than 3 control bits are required for analog memory when weights are trimmed only in inference. The minimum control bit requirement increases to over 5 control bits when weights are adjusted in training and inference because lower control bit precision during training results in less accurate gradient calculations and weights, leading to reduced inference accuracy.

Figure 5b, c analyzes the impact of noise sources on inference accuracy after training without noise and with noise, respectively. The main noise sources that have been taken into account come from the laser, SOA, PD, and TIA. Results indicate that inference accuracy degrades with more than 12.5% optical power noise injected into the weight banks. Furthermore, networks trained with noise are more resilient to PD noise current during inference than networks trained without noise. Specifically, the network trained with noise maintains

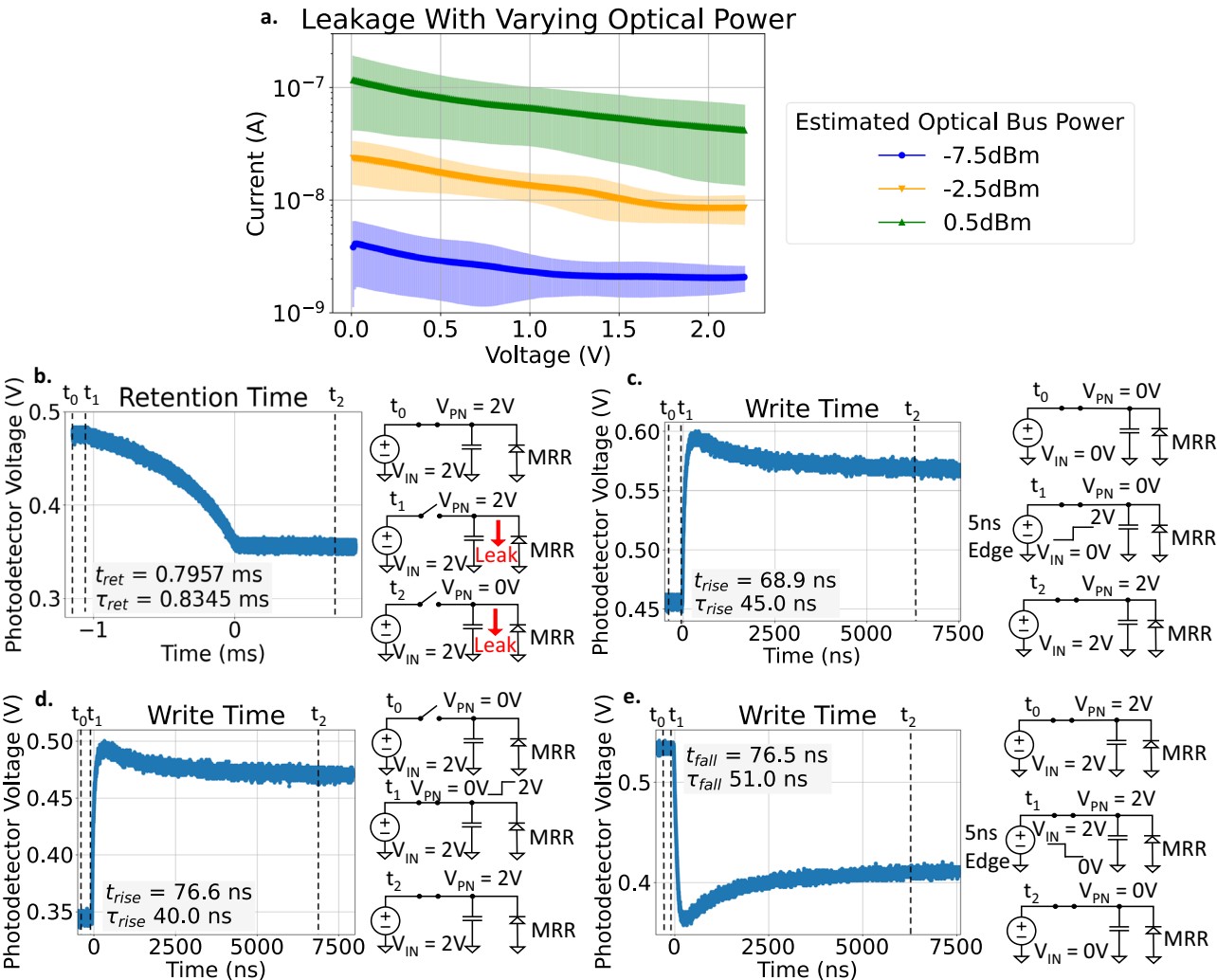

**Fig. 3 | Electro-optic characteristics of the DEOAM. a** Leakage shows a dependence on optical power incident on the doped MRR. Error bands represent the standard deviation in the leakage. **b** Retention time from leakage is 0.835 ms for one time constant ($\tau_{ret}$) at 0.5 dBm optical bus power. The decay shows a nonlinear response from the ring's Lorentzian response. **c** With the transmission gate on, 2 V is written to analog memory using a 5 ns edge rate, and the resulting write time is 45.0 ns for one time constant ($\tau_{rise}$). **d** With a source voltage of 2 V on $V_{IN}$ and toggling the transmission gate from off to on, write time is 40.0 ns ($\tau_{rise}$). **e** With the transmission gate on and writing 0 V to analog memory at an edge rate of 5 ns, write time is 51.0 ns for one time constant ($\tau_{fall}$). Further discussions on the electro-optic results are expanded in Supplementary Information.

high inference accuracy until the PD noise current reaches about 2 mA RMS, whereas the network trained without noise only maintains high inference accuracy until about 0.1 mA RMS of PD noise current.

As weights decay due to the analog memory's leakage, the memory's retention time constant and the network latency determine the number of images that can be classified correctly or the number of useful computations before the weights become unusable. Network latency represents the time for an image to pass through the network. Within a retention time constant, $k$ number of images can be passed through the network before weights are unusable, incurring $k$ times the network latency. This indicates that the ratio of retention time constant to network latency is crucial for a network's inference performance.

Figure 5d examines the impact of analog memory retention time constant (weight decay) and network latency on inference accuracy when it degrades by 10%. The weight decay is modeled as an exponential decay based on the ratio of retention time constant to network latency. These experiments compare scenarios where weights are trained with and without considering retention time and network latency. The findings suggest that training with retention time constant

and network latency relaxes the requirements for analog memory retention time, enabling the use of memories with shorter retention time. A ratio of retention time constant to network latency greater than 100 prevents inference accuracy from decaying by 10%. This implies that even with an analog memory retention time constant of 100 µs, if the network latency is less than 1 µs, the network can maintain high inference accuracy. The ratio of retention time constant to network latency can be scaled to any analog memory technology, giving either a minimum retention time constant for a given network latency or a maximum network latency for a given retention time constant. Although analog memory leakage may limit inference accuracy, controlled leakage can be employed as a regularization technique to avoid overfitting, enabling broader inference applicability and metalearning[50,51]. Furthermore, retention time constant and network latency are varied to evaluate training accuracy with different batch sizes, including 16, 32, and 64. The results show that larger batch sizes require longer retention times to achieve similar training accuracy as smaller batch sizes. With smaller batch sizes, weights are updated more frequently, so the analog memory does not need to retain the data for a long time and can operate with shorter retention times.

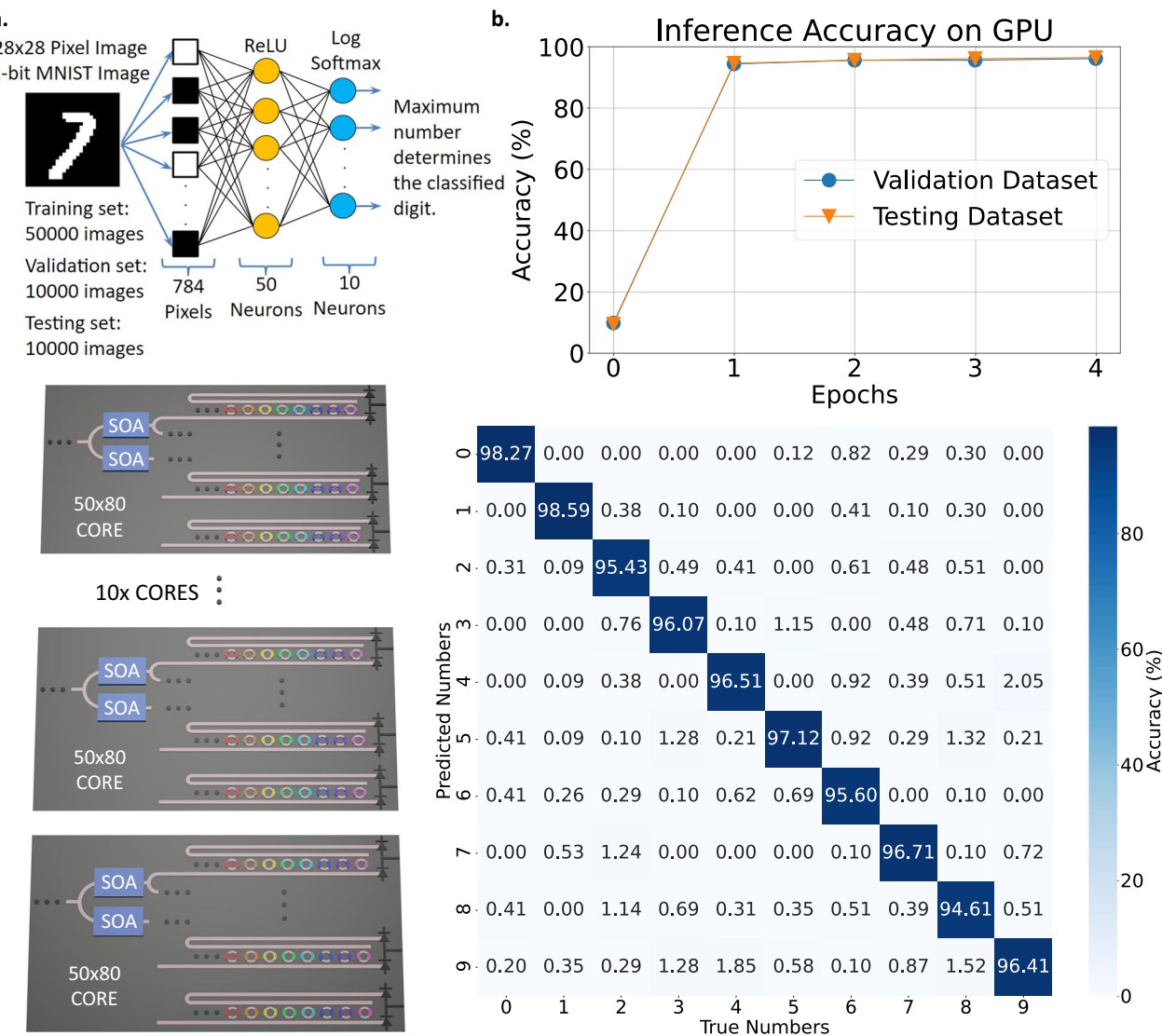

**Fig. 4 | The neural network emulation architecture and initial training. a** The feedforward neural network is trained using the MNIST dataset with 50,000, 1-bit, 28 × 28 pixel images using a batch size of 64, then validated and tested using 10,000 images[21]. The neural network is three layers with the input layer supporting 784 values for the 28 × 28 pixel images, the hidden layer supporting 50 neurons using a ReLU activation function, and the output layer supporting 10 neurons using a logarithmic softmax activation function to classify as a number. The neural network can map to the proposed neuromorphic photonic hardware that consists of 10 photonic cores with 50 rows of weight banks and 80 MRRs in each weight bank. Each core uses 80 wavelengths and 50 semiconductor optical amplifiers (SOA) to compensate for the splitting loss. **b** The neural network achieves more than 95% inference accuracy after one epoch for all numbers. The specifics of the neural network architecture and modeling are detailed in "Methods".

Once weights become unusable due to leakage, dead time is consumed to refresh the weight, which consists of the time needed to update the DAC inputs, analog memory write time, and DAC settling time. For DEOAM, the write time is about 65 ns in the fastest case. Surveyed DACs, in similar process nodes to DEOAM, can achieve sample rates up to 3 GS/s (or 333 ps per sample), and reported SRAM propagation delays are in the 10s of picoseconds, meaning that the time needed to update the DAC inputs and DAC settling time is negligible compared to the DEOAM write time[24,25]. Even for the fastest write time among surveyed analog memory technologies (500 ps for PCMs), the analog memory write time is similar to DAC sample rates[52]. Therefore, the dead time can be approximated by the analog memory's write time. When examining the ratio of retention time to write time for analog memories, this ratio (10,000 for DEOAM) is a couple of orders of magnitude greater than the retention time-to-network-latency ratio (100) required for high inference accuracy. This means the dead-time for refreshing can be much shorter than network latency. Therefore, refreshes can occur after each image classification, thereby maintaining accurate weights for high inference accuracy without adding significant latency, but at the cost of refresh power.

## Discussion

Training neuromorphic photonic processors presents several challenges related to energy efficiency, time, and execution. Analog-to-digital and digital-to-analog conversions, along with memory access, often become energy and latency bottlenecks. Additionally, since modeling hardware and environmental fluctuations are challenging, relying on offline-trained weights can degrade inference accuracy[13,15,53].

To address these challenges, co-locating analog memory with the compute unit can alleviate energy consumption and latency by reducing data movement and the need for ADCs and DACs. Recurrent neural networks, in particular, can benefit from analog memory

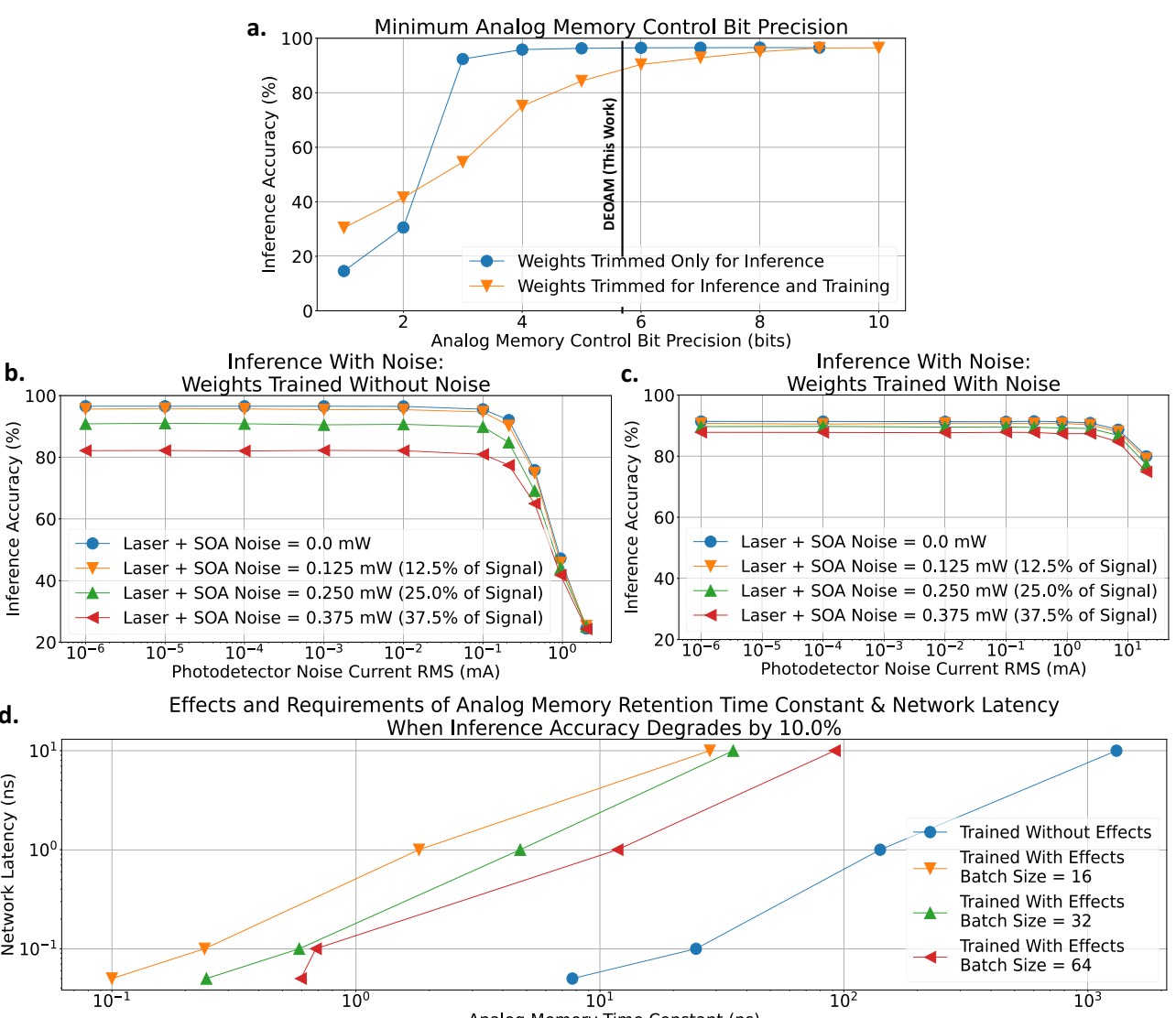

**Fig. 5 | Neural network emulation results with and without non-idealities from the analog memory and system, such as control bit precision, noise, leakage, and latency.** **a** Minimum analog memory control bit precision to achieve more than 95% inference accuracy is 4 bits for weights trimmed only in inference and 8 bits for weights trimmed in inference and training. Inference accuracy degrades with increasing laser, SOA, PD, and TIA noise, but inference accuracy degrades **b** earlier when weights are trained without noise and **c** later when weights are trained with noise. **d** The sweep of network latency and retention time characterizes inference accuracy for weights trained with and without leaky analog memory and network latency. Network latency and analog memory retention time constant at the point when inference accuracy degrades by 10% are plotted. When weights are trained with the effects of network latency and memory retention time, the network compensates for these effects and allows for a lower analog memory retention time constant requirement. Batch sizes of 64, 32, and 16 reveal that large batch sizes require longer retention times since weights are updated less frequently. Weights are updated more frequently in smaller batch sizes. The Supplementary Information expands on results in (**d**).

integration since retaining data in the analog domain reduces analog-to-digital and digital-to-analog conversions[20]. Furthermore, analog memory can enhance the effectiveness of other training algorithms, such as direct-feedback alignment and in-situ training, by improving training time and on-chip computations[15,54]. A potential outcome from integrating analog memory in neuromorphic photonic processors is on-chip, online training, where the processor can adapt to new data coming in, be more resilient to noise and hardware flaws, and consume less energy[53]. Architecturally, incorporating analog memory into neuromorphic photonic processors facilitates system-level improvements, leading to improved energy efficiency, shorter training time, and a more adaptive neural network.

However, scaling the neuromorphic photonic processor with analog memory presents challenges in analog memory and MRR control. As the network scales, reasonable sizes of the network can be up to 108 rings in a weight bank (limited by the ring's FSR and finesse) and up to a few hundred rows of weight banks (limited by SNR)[48,49]. The size of the network can also be limited to the analog memory's retention time, refresh time, and inference time (network latency) where a retention time must be allocated to refresh time for all rows of the network and inference time to ensure the network can operate. For DEOAM, the retention time to write time ratio is about 10,000, and inference times (picosecond to nanosecond range) are negligible compared to the write time. Therefore, if SNR is not limiting the number of rows in the network, the next limitation is the refresh rate, which is limited to several thousand rows. Depending on the analog memory requirements, specialized circuitry may be necessary for writing to or refreshing the memory. As the system scales, strategies

like sharing drive lines (as illustrated in Fig. 1a) where DACs are shared along columns, can be essential in reducing power consumption. Accurate resonance control in MRRs is vital for maintaining precise optical weight representation. As the processor scales, implementing thermal control and multi-MRR control algorithms becomes increasingly critical for system stability[33,46,55–58].

While analog memory can provide system-level enhancements, analog memory specifications must be examined carefully to understand their limitations. Analog memory endurance is particularly critical, as it must endure numerous writes during training. Selecting training algorithms to minimize writes can enhance endurance, but choosing the right analog memory technology for the application is crucial for determining network endurance. Despite DEOAM's leakage, high inference accuracy can still be achieved if the ratio of retention time-to-network latency is 100 or higher. This and other charge-based or leaky analog memories can remain effective and maintain high inference accuracy despite inherent leakage.

The modulator is another area for enhancement. Employing PN junctions with different doping levels and junction geometries, such as interleaved, U-shaped, or L-shaped designs, can improve tuning efficiency and speed[56,59,60]. However, as indicated in Fig. 3a, the PN junction MRR exhibits leakage that is dependent on the incident optical power, suggesting that the photoresponsivity of the PN junction can limit analog memory performance. Reducing this leakage may involve removing contaminants and impurities, such as layering poly-silicon on the substrate or optimizing convection annealing[61,62]. Alternatively, other modulators, such as MOS capacitor modulators, can be explored to potentially reduce leakage[63]. Pockels effect modulators might offer advantages over PN junctions, as they exhibit minimal leakage (less than 100 nA) and better tuning efficiency[64–66]. Further development in foundry processes is needed to better integrate these modulators into electro-optic platforms[67].

We demonstrated a proof-of-concept neuromorphic photonic circuit with DEOAM and verified its capabilities in a neuromorphic system. DEOAM reduces DACs in a neuromorphic photonic processor from $n^2$ to $n$, thereby alleviating bottlenecks in bandwidth and energy consumption. By integrating analog memory on chip, neuromorphic photonic processors have the potential to achieve fast and efficient training. Since the training process occurs on chip, the neural network can readily adapt to environmental fluctuations and noise by incorporating them during training. Moreover, analog memory with on-chip backpropagation hardware can enable online training, allowing weights to be updated after each new input data. This opens up possibilities for various applications requiring recurrent events, event-based decision-making, or adaptability, such as time series forecasting. As a result, the next generation of AI processors, equipped with integrated analog memory, can help sustain the continued growth of AI.

## Methods
### Hardware
In Fig. 2a, the silicon photonics electronic chip, fabricated in the 90 nm GF9WG monolithic process, consists of a $1 \times 4$ MRR circuit with C-band grating couplers to couple light to the chip and electronics to control when to load or reset voltages. The electronics consist of transmission gates and reset (RST) transistors. The transmission gates are controlled by an active low enable (EN) signal, which connects (EN = 0) or disconnects (EN = 1) an external source measure unit (SMU) to the analog memory cells. The RST transistors reset the voltage on the analog memory cells before operation. From Fig. 2c, the silicon photonic electronic chip is thermally glued and wirebonded to a custom printed circuit board (PCB) that breaks out connections to measurement equipment. The PCB has 50 Ω transmission lines that breakout to subminiature version A (SMA) connectors for high speed signals and 2.54 mm pitch headers for power and low-speed signals. A 10k negative temperature coefficient (NTC) thermistor and peltier module are

connected to the LDC501 for thermal control. An 8-degree, 8-channel fiber array is used to couple light to the chip.

Figure 2b shows the operation of the circuit. Initially, the laser is aligned with the MRR resonance while the MRR is voltage-biased at zero voltage. At time 1, the EN signal is active, allowing voltages from the SMU to propagate to the analog memory. At time 2, the SMU drives the capacitor with an analog voltage, which also drives the PN junction MRR. At time 3, when the EN signal is inactive, the capacitor holds the analog voltage on the PN junction MRR until the charge leaks away. As a result, the optical signals on the through (THRU) and drop (DROP) ports of the MRR circuit track the voltage on the capacitor and PN junction MRR.

### Measurement setup
From Fig. 2a, the optical characteristics are measured using a C-band tunable laser (HP81682A) and an array of photodetectors (N7744A) while the MRRs are voltage-biased using an SMU (Keithley 2604B). Light is coupled through an 8-degree, 8-channel fiber array to C-band grating couplers on chip, and the SMU sweeps voltage to observe MRR resonance changes with varying voltage biases.

For the retention time measurement setup, the HMP4040 power supply unit powers the electronics, the LDC501 thermoelectric cooler (TEC) regulates the chip temperature, and the HP81682A tunable C-band laser provides optical power to the circuit for optical processing. To drive signals, the Keithley 2604B SMU is used to drive the analog memory and PN junction MRR and the Agilent 33250A arbitrary waveform generator (AWG) drives the EN signal with a 5 ns edge rate step to quickly connect and disconnect the SMU from the analog memory and PN junction MRR. The DROP port is connected to a PD on the N7744A to tune the laser wavelength to the MRR resonance, and the THRU port is connected to the high-speed, amplified PDA255 and DSO81304A oscilloscope to measure the retention time of the circuit. The retention time measurement sequence is shown in Fig. 3b. At $t_0$, EN is active, and the SMU sets the desired voltage on the analog memory and PN junction MRR. At $t_1$, EN becomes inactive, and the analog memory retains the data on the PN junction MRR. After $t_1$, charge leaks away from the analog memory, which returns the optical response on the THRU and DROP ports back to its original state (back to the MRR's resonance). At $t_2$, all charge is depleted on the analog memory and the optical response returns to the MRR's resonance. The leakage measurement setup is the same as the retention time setup, except that the EN signal is constantly active and the SMU is sweeping the voltage from 0 to 2 V while measuring leakage current through the circuit.

The write time measurement setup is similar to the retention time and leakage measurement setup, except that the AWG is driving the analog memory and PN junction MRR while the SMU is holding the EN signal active. The write time measurement sequence is shown in Fig. 3c, e. The AWG sends a pulse train with 5 ns edge rates to drive the analog memory and PN junction MRR, which modulates the optical response on the THRU and DROP port. The optical response is detected using the THRU port by the PD255 and measured by the DSO81304A oscilloscope. The write time measurement sequence shown in Fig. 3d differs from Fig. 3c, e because the rather than driving the analog memory and PN junction with the 5 ns edge rate pulse train, the transmission gate is driven with a pulse train that connects and disconnects the voltage source from the analog memory and PN junction. The rise time, fall time, and one time constant values are extracted from the oscilloscope time domain response.

### Neural network architecture
To map the neural network onto hardware, the input data consists of $28 \times 28$ 1-bit images (784 pixels), which are mapped to 80 input data modulators per core across 10 cores, for a total of 800 modulators, each handling a single pixel (with some leftover). Each pixel (input

data point) is multiplied by 50 different weights to connect to the hidden layer. Since each core contains 50 rows of weight banks, there are 50 MRRs or weights associated with each pixel. Once an input pixel is multiplied by a weight, summation occurs at the PD within each core and then across cores to obtain the 50 neurons in the hidden layer. In each core, 80 pixels are processed (multiplied by their weights) and summed at the PD in each row of weight banks. To process all 784 pixels for the hidden layer, the summation from a given row in one core is combined with the corresponding row in the other nine cores. This yields the total weighted sum for that row, effectively multiplying 50 weights by each of the 784 pixels to produce the inputs to the 50 hidden neurons. A ReLu activation function is assumed to be implemented in analog electronics. The resulting 50 neurons generate 50 intermediate signals, which are passed to the input data modulators for processing in the output layer. In this stage, each signal is multiplied by 10 weights (10 MRRs), allowing the entire multiplication and summation to fit within a single core. After summation, a logarithmic softmax activation (also assumed to be implemented in analog electronics) classifies the signals to predict the corresponding digit.

## Noise

Laser noise represents input data noise and can be related to relative intensity noise (RIN) from the laser as shown in the following equations[68]:

$$x_i = \overline{x_i}(1 + dx_i) = \overline{P_i}(1 + dP_i) = \overline{P_i}(1 + \mathcal{N}(0, 10^{RIN/10} \cdot f)) \quad (3)$$

where $x_i$ is the $i$th input data, $\overline{x_i}$ is the average value of the $i$th input data, $dx_i$ is the $i$th random noise for the $i$th input data, $\overline{P_i}$ is the $i$th average laser power, $dP_i$ is the $i$th random noise in the laser power, $\mathcal{N}$ is the normal distribution function, $RIN$ is the $i$th relative intensity noise in dB/Hz, and $f$ is the integration bandwidth in Hz. As laser light splits to reach each weight bank, splitting loss is incurred. SOAs can amplify the signal to compensate for splitting loss but adds noise to the input data. As a result, the input data to each weight can be described by:

$$x'_i = x_i + n_{SOA_j} \quad (4)$$

where $x'_i$ is the $i$th input data to the $i$th weight that includes SOA and laser noise, and $n_{SOA_j}$ is the $j$th row SOA noise. As the input data is multiplied by weights, insertion loss from MRRs degrades the input data signal. Weight bank insertion loss ($\beta$) is modeled by using an insertion loss of 0.125 dB/MRR, giving a total insertion loss of 10 dB in each weight bank. At the input of each weight bank, each wavelength has 1 mW of optical power and the noise ranges from 0% to 37.5% of 1 mW to mimic noise from the laser and SOAs. Furthermore, noise is applied to the data after the MAC operation and before the activation function to mimic the input-referred noise current on the PD and TIA. The PD and TIA integrated noise current ranges from 1 nA to 2 mA based on current PD and TIA technology[40–44,69–72]. The following equation summarizes the noise sources and the neural network equation:

$$y_j = \sigma_j(\beta \mathbf{x}' \cdot \mathbf{w}_j + n_j) \quad (5)$$

where $y_j$ is the $j$th output data in a layer, $\beta$ is the weight bank insertion loss, $\sigma_j$ is the $j$th activation function in a layer, $\mathbf{x}'$ is the input data vector to the weight banks that includes SOA and laser noise ($\mathbf{x}' = <x'_0, x'_1, \ldots, x'_i>$), $\mathbf{w}_j$ is the $j$th vector of weights in a layer, and $n_j$ is the $j$th input referred noise term. The dot product of $\mathbf{x}'$ and $\mathbf{w}_j$ represents the photocurrent from the PD detecting laser wavelengths weighted by MRRs.

## Weight decay and network latency

For each weight, weight decay is modeled by using a ratio of retention time to network latency in an exponential decay model to determine inference accuracy from leaky weights:

$$w(k) = w_0 e^{-k\frac{t_{latency}}{\tau_{ret}}} \quad (6)$$

where $w(k)$ is the weight value for the $k$th image, $w_0$ is the initial weight, $t_{latency}$ is the network latency or compute time, $\tau_{ret}$ is the retention time constant. In inference, network latency is determined by group delay from photons propagating through the optical circuit and driver, modulator, PD and TIA bandwidth, whereas training has additional latency from analog electronics used to update weights. Since the bandwidth of the PD and TIA and analog electronics can vary depending on design and technology nodes, the range of network latencies is assumed to be determined by the PD and TIA bandwidth, which translates to latencies ranging from 50 ps to 10 ns[40–44,69–71].

## Data availability

Methods, source data, experimental setup, and additional analyses and results are provided in the Supplementary Information and are available from the corresponding authors on request. The measurement data and source code in this study have been deposited in the Figshare database under the accession code https://doi.org/10.6084/m9.figshare.30176863.

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

## Acknowledgements

We gratefully acknowledge Vikas Vijay Kumar and the GlobalFoundries University Program team for their generous MPW support, and we acknowledge the Natural Sciences and Engineering Research Council of Canada and CMC Microsystems for their funding support and design tools. We also acknowledge Omid Esmaeeli and Mohammed Al-Qadasi for helping with design reviews, design rule checks, and tapeout; Mustafa Hammood for helping with wirebonding; Jonathan Barnes and Madeline Mahanloo for helping with measurements; Avilash Mukherjee, Hassan Talaeian, Ata Khorami, Mohammed Al-Qadasi, and Ben Cohen for reviewing this paper. S.L., S.S., L.C., and B.J.S. acknowledge support from the Natural Sciences and Engineering Research Council of Canada (NSERC). B.J.S. is supported by the Canada Research Chairs Program and the Sloan Foundation. S.S. is supported by the Schmidt Science Polymath Award. S.B. acknowledges funding from the Fonds de recherche du Québec-Nature et technologies.

## Author contributions

S.L., S.S., B.A.M., and B.J.S. conceived the idea. S.L. performed the experiments, led the manuscript writing. S.L., S.S., and S.B. architected, and S.L. and S.B. designed the chip. A.K. and S.L. developed and analyzed the simulation studies. S.L., A.K., S.B., B.A.M., P.R.P., L.C., B.J.S., and S.S. analyzed and discussed the data and contributed to revising the manuscript. S.S. and B.J.S. co-supervised this study.

## Competing interests

B.J.S. and B.A.M. cofounded Milkshake Technology Inc. The remaining authors declare no competing interests.
