## [Transparent Peer Review file · Nature Communications]

Neuromorphic Photonic Computing with an Electro-Optic Analog Memory

Corresponding Author: Mr Sean Lam

Version 0:

Reviewer comments:

Reviewer #1

(Remarks to the Author)

In this paper, the authors present an analog memory system integrated directly with photonic computing devices. Their approach offers improved efficiency by eliminating the need for data transfer between memory and converters. It is fully CMOS-compatible, and the experimental results are solid. However, a few clarifications—particularly regarding comparisons to the current state of the art—are necessary before I can recommend this work for publication in Nature Communications.

1. Scalability: What are the specific advantages of using light in this context? While optics can offer low latency, I acknowledge, they tend to be bulkier than electronic components. In your view, what is the key advantage of optical neuromorphic computing compared to the most advanced purely electronic implementations?
2. State-of-the-Art (Photonics): I believe several key papers in the field of optical neuromorphic computing should be cited and discussed. Notably, the work by Feldmann et al., "Parallel convolutional processing using an integrated photonic tensor core," *Nature* 589, 52–58 (2021), <https://doi.org/10.1038/s41586-020-03070-1>, along with its references and selected citations, should be included. In particular, the area of optical neuromorphic memristors is underrepresented in the current discussion and, in my view, merits more thorough coverage.
3. A recent paper by Weilenmann, C., Ziogas, A.N., Zellweger, T. et al., "Single neuromorphic memristor closely emulates multiple synaptic mechanisms for energy-efficient neural networks," *Nat Commun* 15, 6898 (2024), <https://doi.org/10.1038/s41467-024-51093-3>, presents a nanoscale two-terminal device capable of exhibiting both short- and long-term plasticity, with weight decay constants ranging from milliseconds to seconds—achieved without relying on large capacitors. The authors effectively leverage the intrinsic device dynamics in a recurrent neural network (RNN) application. This work could serve as a source of inspiration, particularly in relation to the leaky dynamics discussed in your capacitive memory (see lines 172–174 and the Discussion section).
4. Latency and Energy-Delay Product (EDP): A theoretical analysis of the latency and EDP for the complete electro-optical system—including the photodetector (PD), laser, modulator, semiconductor optical amplifier (SOA), and transimpedance amplifier (TIA)—would be valuable. This would provide an estimate of the potential gains from adopting a fully optical solution. It might also be helpful to include latency and EDP values in Table F2 for comparison.
5. Figure 3a and 3b: What is the physical origin of optical-induced extension of the retention times?

Overall, I appreciate the concept and execution of this work. It is a strong and solid contribution that I believe deserves publication in Nature Communications, provided the points raised above are addressed.

Reviewer #2

(Remarks to the Author)

This manuscript demonstrates an electro-optic analog short-term memory (< 1 ms retention time). The integrated device features a reserved biased PN-type microring resonator (MRR) driven by a capacitor. Due to leakage, this configuration is naturally limited by a short retention time. Based on the characterization results of a single device, emulation of neural network implementation is carried out to analyze the potential performance metrics under various assumed conditions. While the effort to monolithically integrate CMOS-compatible memory with photonic integrated circuit is a meaning route to explore, the manuscript falls short in providing only limited experimental results, which are not able to concretely support the major claims of the manuscript. The key experimental results are only presented in Fig.3 and Fig.2C, while the rest are simulation

or are conceptual. No actual computing experiment is implemented and demonstrated.

Major comments are:

1. The claim of energy reduction in Equation (2) is confusing. It seems Equation (2) can only account for the power consumption of writing one row of MRR. In order to write all n rows, the overall power consumption still scales with n^2 . For the SRAM-DAC architecture, if a capacitor is added after the DAC, would the overall power consumption be similar to DEOAM, except that some footprints are reduced?
2. Fast modulation is desired for training and long retention time is desired for inference. The proposed device is somewhere in between with moderate modulation speed and short retention time. Given the analog nature of photonic computing, such short-retention time combined with moderate modulation speed does not look sensible, as bit error rate will certainly deteriorate.
3. The capability of online training is only supported by simulation.

Minor comments are:

- a. The capacitance of MRR should be considered when determining the lower limit of memristor capacitance (is 100 fF in Fig.1 feasible?). And the possible higher leakage should also be considered when the memristor capacitance is reduced.
- b. May the authors clarify 'DAC can be reused'? Does it mean the same DAC will be used row by row? Or one DAC in column m can program all MRRs in column m in one time step?
- c. It is unclear how the 5-bit resolution is obtained from Figure 3.
- d. The discussion about Fig.4 should be strengthened. Now there is only one sentence describing the whole of Fig.4.
- e. How to use the 10 cores of 50 rows of weight banks each containing 80 MRRs to run the neural network in Fig.4a should be elaborated.
- f. May the authors present data showing >8000 cycles endurance?
- g. Regarding the crosstalk characterization in Appendix B and Fig. C2, the results indicate that the wavelength shift magnitudes between peak 3 and peaks 1,2, and 4 are comparable but occur in opposite directions. Could authors explain why the peaks shift in opposite directions? Furthermore, it appears that the wavelength shift magnitude is not negligible when only tuning one ring resonator.

Reviewer #3

(Remarks to the Author)

Reviewer #4

(Remarks to the Author)

The authors present a new approach to neuromorphic photonic computing, employing analog memory co-located with photonic devices. The paper is technically sound, well organized and well written, but the degree of novelty is moderate in my opinion. Here I have a list of comments that can help the authors improve their manuscript:

- The proposed approach consists of employing a capacitance to store the voltage needed to tune the MRR resonator. In this way, in an " $n \times n$ " matrix, instead of employing n^2 DACs, only n DACs – one for each column – are used, saving power, area and complexity. This approach of employing time-multiplexed DACs followed by hold-capacitors (very popular in standard electronic applications) presents some drawbacks in terms of ratio between retention time and network latency, as the authors highlighted. However, I would appreciate more insights about that, because there are several aspects neglected in this work that can become dominant. The dead-time for refreshing is not only the write time of the single capacitor multiplied the number of capacitors. Also the settling time of the DAC, needed to change its output voltage, could be relevant, as well as the time needed to update the input of the DAC (e.g. the write time of the SRAM preceding the DAC). Please clarify this point.
- Analogously, in equation (2) - concerning the power consumed by DEOAM - some terms are missing. In the standard approach, the power consumed by each SRAM-DAC is only static, and depending on the kind of DAC, can be very low. With the DEOAM approach, instead, the dynamic power of the DAC can be a relevant portion of the DAC power. Moreover, in DEOAM also the inputs of the DAC are dynamically changed, so also this contribution needs to be taken into account.
- The abstract of the manuscript is only qualitative. Some quantitative results would be appreciated.
- I recommend including the details of the capacitance implementation, namely the metal-oxide-metal technology and dimensions, within the main body of the work rather than relegating this information solely to the appendix. Also, the size of the PN-junction MRR should be added in the text, to understand the possibility of scaling up the size of the neural networks.
- Related to the previous point, it would be helpful to add some considerations about the scalability of the proposed approach with larger networks. Considering the trade-off between refresh time and inference time of the network, what are the reasonable maximum sizes (with the relative area consumption) achievable with this approach?
- Even if the front-end was not integrated in the prototype, it is important to give more details about that because it also impacts on the overall performance, specially in terms of latency of the network.
- In the Discussion section, the authors mention "backpropagation hardware", but there are no details or references about that.

Version 1:

Reviewer comments:

Reviewer #1

(Remarks to the Author)

My comments have been addressed.

Reviewer #2

(Remarks to the Author)

The authors have addressed all my concerns. I would recommend publication, and hope to see future excellent works on advanced system functionality.

Reviewer #3

(Remarks to the Author)

Reviewer #4

(Remarks to the Author)

The authors have addressed all the points raised in the previous revision round and improved their manuscript accordingly. I do not have further questions.

Point-by-Point Response to Reviewers' Comments

Blue - Authors' Response

Red - Manuscript Edits

Reviewer #1 (Remarks to the Author):

In this paper, the authors present an analog memory system integrated directly with photonic computing devices. Their approach offers improved efficiency by eliminating the need for data transfer between memory and converters. It is fully CMOS-compatible, and the experimental results are solid. However, a few clarifications—particularly regarding comparisons to the current state of the art—are necessary before I can recommend this work for publication in Nature Communications.

Thanks. We have addressed your comments below to incorporate more clarifications to the manuscript.

1. Scalability: What are the specific advantages of using light in this context? While optics can offer low latency, I acknowledge, they tend to be bulkier than electronic components. In your view, what is the key advantage of optical neuromorphic computing compared to the most advanced purely electronic implementations?

- (1) Neuromorphic photonic processors can exploit multiple degrees of freedom of light to enable parallel, high-bandwidth computations. These include wavelength-division multiplexing [Feldmann et al., Nature 589, 52–58 (2021)], mode multiplexing [Khaled et al., arXiv:2411.15339, Nat. Commun. (accepted)], time-domain multiplexing [Lin et al., Nat. Commun. 15, 9081 (2024)], and hybrid approaches combining these techniques [Ou et al., Sci. Adv. 11, eadu0228 (2025)]. Whereas in electronics, electrical interconnects can only carry a single high-speed baseband signal or multiple narrow-band frequency-division-multiplexed signals and are constrained by crosstalk when parallelizing to multiple electrical interconnects or frequency bins.
- (2) Optical interconnects do not suffer from limitations such as inductance, capacitance or skin effects, allowing for low-latency, energy efficient processing. As a result, photonic processors offer the potential to significantly outperform their electronic counterparts.

We have added:

40 extend the domain of AI [5–8]. The key advantages of neuromorphic photonic accelerators over purely electronic
41 implementations include 1) massive parallelism using optical techniques, such as wavelength-division multiplexing
42 (WDM), mode multiplexing (MDM), and time-domain multiplexing to carry multiple signals in a single inter-
43 connect and 2) lower energy-delay products in optical interconnects for multiply-accumulate (MAC) operations
44 with potential speeds on the order of peta MACs per second per mm² and energy efficiencies in the atto joule
45 per MAC regime [5, 9–12]. However, most photonic processors have been limited to stationary weights, which

2. State-of-the-Art (Photonics): I believe several key papers in the field of optical neuromorphic computing should be cited and discussed. Notably, the work by Feldmann et al., "Parallel convolutional processing using an integrated photonic tensor core," Nature 589, 52–58 (2021), <https://doi.org/10.1038/s41586-020-03070-1>, along with its references and selected citations, should be included. In particular, the area of optical neuromorphic memristors is underrepresented in the current discussion and, in my view, merits more thorough coverage.

Thank you for the suggestion. We have added the following lines:

58 [25, 26]. Phase change materials (PCM) have been demonstrated in a photonic tensor core enabling 10^{12} MACs
 59 per second for inference tasks [9]. However, the endurance of PCMs can vary depending on the modulation mech-
 60 anism (electrical, electrothermal, or optical), raising concerns about their suitability for neuromorphic training
 61 applications that typically require volatile memory [27]. Optical memristors, which store memory by forming and
 62 dissolving filaments, offer minimal footprint, making them promising candidates for large scale integration. How-
 63 ever, their limited endurance (approximately 2,000 write cycles) and challenges in compatibility with standard
 64 foundry processes remain key obstacles to broader adoption [25, 28]. In contrast, analog electronic memories, such

3. A recent paper by Weilenmann, C., Ziogas, A.N., Zellweger, T. et al., "Single neuromorphic memristor closely emulates multiple synaptic mechanisms for energy-efficient neural networks," Nat Commun 15, 6898 (2024), <https://doi.org/10.1038/s41467-024-51093-3>, presents a nanoscale two-terminal device capable of exhibiting both short- and long-term plasticity, with weight decay constants ranging from milliseconds to seconds—achieved without relying on large capacitors. The authors effectively leverage the intrinsic device dynamics in a recurrent neural network (RNN) application. This work could serve as a source of inspiration, particularly in relation to the leaky dynamics discussed in your capacitive memory (see lines 172–174 and the Discussion section).

Thank you for the inspiration. We have added this reference to our manuscript:

196 latency for a given retention time constant. Although analog memory leakage may limit inference accuracy,
 197 controlled leakage can be employed as a regularization technique to avoid overfitting, enabling broader inference
 198 applicability and meta-learning [51, 52]. Furthermore, retention time constant and network latency are varied

4. Latency and Energy-Delay Product (EDP): A theoretical analysis of the latency and EDP for the complete electro-optical system—including the photodetector (PD), laser, modulator, semiconductor optical amplifier (SOA), and transimpedance amplifier (TIA)—would be valuable. This would provide an estimate of the potential gains from adopting a fully optical solution. It might also be helpful to include latency and EDP values in Table F2 for comparison.

Specification	Laser [82]	SOA [83]	DEOAM (This Work)	PD and TIA [79]	Input Data Modulators [84]	Thermal Stabilizers [85]
Average Power Consumption Per Device (mW)	100 - 1000	300 - 1000	1 ¹	37	-	<30
Energy Consumption Per Device (μ J)	5 - 50	15 - 50	0.05	1.85	40E-9 - 1E-6	<1.5
Required Number of Devices	10	50x10 = 500	50x80x10 = 40000	50x10 = 500	80x10 = 800	50x80x10 + 10 = 40010
Total Energy Consumption (μ J)	50 - 500	7500 - 25000	2000	925	32E-6 - 800E-6	<60015

Table F2: Training energy consumption analysis for the neuromorphic photonic circuit in Figure 4 using DEOAM. Total energy consumption for each device group is calculated from average power consumption, training time (for DEOAM which is 50μ s from Figure E3), and number of devices required for the neural network. Results show that optical power delivery (lasers and SOAs), thermal stabilization, and DEOAM consume the most amount of energy.

¹Calculated based on nominal write energy and write time.

Thank you for the suggestion. We have included an EDP analysis as part of the energy consumption analysis. We have added:

- Equations F5, F6, F7, and F8 to outline the equations deriving EDP
- Figure F5 to show the trend in EDP as the network size scales
- Description and discussion:

477 F.5 Energy Delay Product

478 Energy consumption is dependent on the neural network architecture and key devices that consume power, which
 479 include lasers, SOAs, DEOAM, PDs, TIAs, input data modulators, and thermal stabilizers. Table F2 summarizes
 480 the energy consumption of key devices in the neuromorphic photonic circuit from Figure 4 during training.
 481 Table F2 reveals that optical power delivery via lasers and SOAs and thermal stabilizers consume most of the
 482 energy during training followed by DEOAM. Thermal stabilizers using metallic heaters can be replaced by more
 483 efficient phase shifters, thereby reducing the energy consumption significantly [68, 87]. Future research for energy
 484 efficient neuromorphic photonic processors should, therefore, focus on improving optical power delivery (lasers
 485 and SOAs), photonic device stabilization (efficient phase shifters and circuits), and analog memory technology
 486 (materials and circuits).

487 To analyze the power consumption as the network size scales, we consider an $n \times n$ network comprising n
 488 inputs, $n \times n$ weights, and n outputs. In such a configuration, the system includes: n input data modulators,
 489 each with a thermal stabilizer, n SOAs, n PD and TIAs, n^2 MRRs each with a DEOAM and thermal stabilizer,
 490 and lasers that are assumed to be shared among a maximum of 50×50 MRRs. The total power consumption is
 491 then calculated using the device power values listed in Table F2 and the following equation:

$$P_{total} = \text{ceil}\left(\frac{n}{50}\right)P_{laser} + nP_{SOA} + n^2P_{DEOAM} + nP_{PD+TIA} + nP_{input-mod} + (n + n^2)P_{therm} \quad (\text{F5})$$

492 where P_{total} is the total power consumption; ceil is the ceiling function; P_{laser} , P_{SOA} , P_{DEOAM} , P_{PD+TIA} ,
 493 $P_{input-mod}$, and P_{therm} are the single device power consumption values for the laser, SOA, DEOAM, PD and
 494 TIA, input data modulators, and thermal stabilizers, respectively.

495 The inference delay path, t_{delay} , starts from the input data drivers and modulators, passes through waveguides,
 496 splitters, weight banks, and ends at the PD and TIA. Optical delay includes both the optical path length and
 497 the resonance build up time. It is given by:

$$t_{delay} = t_{DRV+MOD} + \frac{n_g}{c}(2nd_{MRR} + FR + L_{splitter}\log_2(n)) + t_{PD+TIA} \quad (\text{F6})$$

498 where $t_{DRV+MOD}$ is the combined propagation delay of the driver and activation modulator (assumed to be data-
 499 rate limited, ranging from 10 Gb/s to 200 Gb/s [90]), n_g is the optical group delay, c is the speed of light, d_{MRR}
 500 is the MRR separation distance, F is the MRR finesse, R is the ring radius, $L_{splitter}$ is the length of a splitter,
 501 and t_{PD+TIA} is the combined propagation delay of the PD and TIA. The energy consumed per operation E_{op} is:

$$E_{op} = \frac{P_{total}t_{delay}}{n^2} \quad (\text{F7})$$

511 estimate. While using the best-in-class devices could yield more optimistic results, integrating all such technologies
 512 remains a significant challenge. The projected trend indicates that network sizes with n ranging from 100 to
 513 1000 achieve minimal EDP. For $n > 1000$, EDP increases due to rising energy consumption and delay. Current
 514 implementations typically exhibit EDP around 10^{-13} or higher. Although present implementations still have
 515 higher EDP, the projections show the potential or substantial improvement if optimal devices can be integrated.
 516 Furthermore, the EDP trends serve both as targets for future development and as indicators of architectural
 517 scalability. This enables neuromorphic hardware designers to assess the feasibility and suitability of neuromorphic
 518 photonic systems for large-scale implementations.

502 where there are n^2 multiply-accumulate (MAC) operations in an $n \times n$ network. The energy–delay product
 503 (EDP) is then:

$$EDP = E_{op} t_{delay} \quad (F8)$$

504 This EDP can be used to compare different neuromorphic processors as shown in Fig. F5. The EDP projection
 505 shown is based on the worse device performance surveyed from Table F2, providing a conservative performance
 506 estimate. While using the best-in-class devices could yield more optimistic results, integrating all such technologies
 507 remains a significant challenge. The projected trend indicates that network sizes with n ranging from 100 to
 508 1000 achieve minimal EDP. For $n > 1000$, EDP increases due to rising energy consumption and delay. Current
 509 implementations typically exhibit EDP around 10^{-13} or higher. Although present implementations still have
 510 higher EDP, the projections show the potential or substantial improvement if optimal devices can be integrated.
 511 Furthermore, the EDP trends serve both as targets for future development and as indicators of architectural
 512 scalability. This enables neuromorphic hardware designers to assess the feasibility and suitability of neuromorphic
 513 photonic systems for large-scale implementations.

Fig. F5: Energy delay product (EDP) as the network scales $n \times n$ (n rows and n columns). The estimated EDP projection uses devices with worse device performance metrics from Table F2. Energy delay product is compared to state-of-the-art neuromorphic photonic processors [92].

5. Figure 3a and 3b: What is the physical origin of optical-induced extension of the retention times?

The PN junction in the ring acts as a photodiode that generates an electrical current dependent on the optical power incident on the PN junction. Higher optical power leads to higher photocurrent, which drains the charge on the capacitor faster, leading to a shorter retention time. This is briefly explained:

¹⁴¹ is effectively zero. Retention time is dependent on leakage, and leakage is dependent on optical power and
¹⁴² photoresponse of the PN junction MRR [45]. Figures 3a and b reveal that the retention time of 0.8345 ms for one

Overall, I appreciate the concept and execution of this work. It is a strong and solid contribution that I believe deserves publication in Nature Communications, provided the points raised above are addressed.

Thank you for your kind comments.

Reviewer #2 (Remarks to the Author):

This manuscript demonstrates an electro-optic analog short-term memory (< 1 ms retention time). The integrated device features a reserved biased PN-type microring resonator (MRR) driven by a capacitor. Due to leakage, this configuration is naturally limited by a short retention time. Based on the characterization results of a single device, emulation of neural network implementation is carried out to analyze the potential performance metrics under various assumed conditions. While the effort to monolithically integrate CMOS-compatible memory with photonic integrated circuit is a meaning route to explore, the manuscript falls short in providing only limited experimental results, which are not able to concretely support the major claims of the manuscript. The key experimental results are only presented in Fig.3 and Fig.2C, while the rest are simulation or are conceptual. No actual computing experiment is implemented and demonstrated.

Thank you for your comment. In this proof-of-concept work, our primary goal was to validate the most innovative aspects of our approach. The design was constrained by the limited chip area gifted to us by our collaborators during a shared wafer run. Additionally, demonstrating a large scale neural network would involve costs that are beyond the scope of a typical academic research group. To illustrate what is possible at scale, we cite recent work from Lightmatter, which showcases what a well-funded startup with a large engineering team can achieve.

[89] Ahmed, S. R. *et al.* Universal photonic artificial intelligence acceleration. *Nature* **640**, 368–374 (2025).

Major comments are:

1. The claim of energy reduction in Equation (2) is confusing. It seems Equation (2) can only account for the power consumption of writing one row of MRR. In order to write all n rows, the overall power consumption still scales with n². For the SRAM-DAC architecture, if a capacitor is added after the DAC, would the overall power consumption be similar to DEOAM, except that some footprints are reduced?

Both the SRAM-DAC and DEOAM memory architectures write to only one row at a time, resulting in equivalent dynamic DAC power consumption. However, their static power differs: in the SRAM-DAC architecture, static DAC and SRAM power consumption scale with n², while in the DEOAM architecture, static DAC power scales linearly with n. To make this distinction clear, we have updated the equations by separating the total DAC power P_{DAC} into P_{DAC_static} and P_{DAC_dynamic}:

$$P_{SRAM-DAC} = n^2(P_{DAC_{static}} + P_{SRAM_{static}}) + n(P_{DAC_{dynamic}} + P_{SRAM_{dynamic}}) \quad (1)$$

$$P_{DEOAM} = n(P_{DAC_{static}} + P_{DAC_{dynamic}} + \alpha C f V^2)$$

From the two equations, the DEOAM architecture consumes less power than the SRAM-DAC architecture when $nP_{SRAM_{dynamic}} + n^2(P_{DAC_{static}} + P_{SRAM_{static}}) > n(P_{DAC_{static}} + \alpha C f V^2)$. Here, the left-hand side of the inequality represents the SRAM-DAC architecture power consumption and the right side of the inequality corresponds to the DEOAM architecture power consumption. Introducing an additional capacitor after the DAC in the SRAM-DAC design adds another $n\alpha C f V^2$ term to the left side of the inequality, further increasing the SRAM-DAC's power consumption—thus making the DEOAM architecture consistently more power-efficient.

2. Fast modulation is desired for training and long retention time is desired for inference. The proposed device is somewhere in between with moderate modulation speed and short retention time. Given the analog nature of photonic computing, such short-retention time combined with moderate modulation speed does not look sensible, as bit error rate will certainly deteriorate.

The ratio of the retention time to latency is a critical factor for both inference and training. While longer retention times are generally beneficial, our current design provides an adequate ratio to support both inference and training. Furthermore, we propose several techniques to improve this ratio in future iterations. In our implementation, the weight update modulation speed is on the order of tens of nanoseconds---*comparable* to DRAM memory access in electronic accelerators. However, by integrating memory directly next to the processing unit, our architecture significantly reduces data movement and associated energy costs. Activations are implemented using PN junction modulators capable of operating at GHz rates. Due to the analog nature of photonic computing, accuracy does not degrade significantly as long as the retention-to-latency ratio exceeds 100, as shown in Fig. F4.

Fig. F4: The sweep of network latency and retention time characterizes inference accuracy for weights trained a) without leaky analog memory and network latency and b) with leaky analog memory and network latency. a) A ratio of latency to retention time of 100-300 is sufficient to achieve more than 95% inference accuracy. b) A ratio of latency to retention time of 100 is sufficient to achieve more than 90% inference accuracy. Batch size of b) 64, c) 32, and d) 16 reveal that large batch sizes require longer retention times since weights are updated less frequently. Weights are updated more frequently in smaller batch sizes.

3. The capability of online training is only supported by simulation.

Thank you for your comment. In this proof-of-concept work, our focus was on validating the most innovative aspects of the DEOAM-based architecture. On-chip online training could not be implemented due to area and resource constraints. Nevertheless, we hope this work serves as a foundation for future research and inspires commercial adoption of this approach.

Minor comments are:

a. The capacitance of MRR should be considered when determining the lower limit of memristor capacitance (is 100 fF in Fig.1 feasible?). And the possible higher leakage should also be considered when the memristor capacitance is reduced.

Indeed, the lower limit on the capacitance is set by the junction capacitance of the PN junction MRR, which is approximately <50 fF for the PN junction length used in our device, as estimated based on the following references:

- Guoliang Li, Xuezhe Zheng, Jin Yao, Hiren Thacker, Ivan Shubin, Ying Luo, Kannan Raj, John E. Cunningham, and Ashok V. Krishnamoorthy, "25Gb/s 1V-driving CMOS ring modulator with integrated thermal tuning," *Opt. Express* 19, 20435-20443 (2011)
- H. Li et al., "A 3-D-Integrated Silicon Photonic Microring-Based 112-Gb/s PAM-4 Transmitter With Nonlinear Equalization and Thermal Control," in *IEEE Journal of Solid-State Circuits*, vol. 56, no. 1, pp. 19-29, Jan. 2021, doi: 10.1109/JSSC.2020.3022851.
- R. Dubé-Demers et al., "Analytical Modeling of Silicon Microring and Microdisk Modulators With Electrical and Optical Dynamics," in *Journal of Lightwave Technology*, vol. 33, no. 20, pp. 4240-4252, 15 Oct. 15, 2015, doi: 10.1109/JLT.2015.2462804.

Leakage in our device stems primarily from the PN junction. Reducing the metal-oxide-metal (MOM) capacitance does not impact the leakage current but does influence the retention time, as a smaller capacitance holds less charge. In the *Discussion* section, we outline several methods to mitigate the leakage of the PN junction, which in turn reduces the required MOM capacitance and improves retention time. We have added the following lines to explicitly highlight the limit on the capacitance.

¹⁰⁷ (Figure 1c, inset). As the DEOAM capacitance decreases, further power savings can be realized up to the limit
¹⁰⁸ of the junction capacitance of the PN junction (< 50 fF [32, 33]). Hence, integrating DEOAM and reducing the

b. May the authors clarify 'DAC can be reused'? Does it mean the same DAC will be used row by row? Or one DAC in column m can program all MRRs in column m in one time step?

One DAC can be used row by row. We have updated the description to clarify this point:

Fig. 1: Motivation for electro-optic processors with dynamic electro-optic analog memory (DEOAM). a) Dynamic electro-optic analog memory (DEOAM) consists of a capacitor connected to a PN junction microring resonator (MRR). The capacitor holds data on the MRR, thereby enacting a weight in the optical domain. Using DEOAM means DACs can be shared amongst columns of analog memory devices, updating them row by row since the analog memory holds the signal on the MRR, which relaxes limitations on energy efficiency and bandwidth imposed by DACs. b) Conventional electro-optic

c. It is unclear how the 5-bit resolution is obtained from Figure 3.

The 5-bit resolution is calculated from the maximum and minimum voltage levels then the standard deviation in the signal due to noise. See lines:

146 that the write time is about 40 to 50 ns for one time constant, enabling MHz write speeds. From Figure 3, the
 147 analog memory bit precision is about 5 bits and can be extracted from the time domain responses based on
 148 $\log_2\left(\frac{\bar{\mu}_{max}-\bar{\mu}_{min}}{\sigma}\right) = N_b$ where $\bar{\mu}_{max}$ and $\bar{\mu}_{min}$ are the mean values at the maximum and minimum range of the
 149 analog memory, respectively, σ is the standard deviation, and N_b is the bit precision of the analog memory [46].

d. The discussion about Fig.4 should be strengthened. Now there is only one sentence describing the whole of Fig.4.

Fig. 4: The neural network emulation architecture and initial training. a) The feedforward neural network is trained using the MNIST dataset with 50000, 1-bit, 28x28 pixel images using a batch size of 64, then validated and tested using 10000 images [27]. The neural network is three layers with the input layer supporting 784 values for the 28x28 pixel images, the hidden layer supporting 50 neurons using a ReLU activation function, and the output layer supporting 10 neurons using a logarithmic softmax activation function to classify as a number. The neural network can map to the proposed neuromorphic photonic hardware that consists of 10 photonic cores with 50 rows of weight banks and 80 MRRs in each weight bank. Each core uses 80 wavelengths and 50 semiconductor optical amplifiers (SOA) to compensate for the splitting loss. b) The neural network achieves more than 95% inference accuracy after one epoch for all numbers. The specifics of the neural network architecture, performance, and modeling are detailed in Appendix F.

We have moved part of Appendix F to this section to discuss the architecture:

150 2.2 Analysis of Analog Memory Specifications

151 To verify different analog memory specifications within a neural network system, a weight bank architecture,
 152 depicted in Figure 4, is emulated to assess analog memory performance during inference and training using the
 153 MNIST dataset [30]. The neural network architecture is configured as a three-layer model. The input layer pro-
 154 cesses 784 values corresponding to the 28×28 pixel image. This is followed by a hidden layer with 50 neurons
 155 and a ReLU activation function, and finally, an output layer with 10 neurons and a logarithmic softmax activa-
 156 tion for digit classification. The network achieves over 95% validation and testing accuracy across all predicted
 157 digits within one training epoch, as shown in Figure 4b, which is comparable to the photonic neural network
 158 demonstrated in [48]. For a 5-bit resolution MRR-based weight bank matrix incorporating SOAs and MRRs with
 159 a finesse of 368 (≈ 55000 quality factor), the network size is limited to 108 rings (wavelength channels) per weight
 160 bank due to the MRRs' FSR and finesse. The number of rows (spatial channels) is limited to 60 due to the
 161 signal-to-noise ratio (SNR) constraints [49, 50]. To ensure compatibility within a single core, the implementation
 162 uses 80 MRRs and 80 analog memories arranged in 50 rows of weight banks. Each weight bank requires two PDs
 163 and one TIA to perform the summation and signal amplification.

e. How to use the 10 cores of 50 rows of weight banks each containing 80 MRRs to run the neural network in Fig.4a should be elaborated.

We have updated the manuscript to explain how to run the neural network in Fig. 4a on the hardware:

433 Appendix F Emulation Modeling

434 F.1 Neural Network Architecture and Performance

435 To map the neural network onto hardware, the input data consists of 28×28 1-bit images (784 pixels), which
 436 are mapped to 80 input data modulators per core across 10 cores, for a total of 800 modulators, each handling
 437 a single pixel (with some leftover). Each pixel (input data point) is multiplied by 50 different weights to connect
 438 to the hidden layer. Since each core contains 50 rows of weight banks, there are 50 MRRs or weights associated
 439 with each pixel. Once an input pixel is multiplied by a weight, summation occurs at the PD within each core and
 440 then across cores to obtain the 50 neurons in the hidden layer. In each core, 80 pixels are processed (multiplied
 441 by their weights) and summed at the PD in each row of weight banks. To process all 784 pixels for the hidden
 442 layer, the summation from a given row in one core is combined with the corresponding row in the other nine
 443 cores. This yields the total weighted sum for that row, effectively multiplying 50 weights by each of the 784 pixels
 444 to produce the inputs to the 50 hidden neurons. A ReLu activation function is assumed to be implemented in
 445 analog electronics. The resulting 50 neurons generate 50 intermediate signals, which are passed to the input data
 446 modulators for processing in the output layer. In this stage, each signal is multiplied by 10 weights (10 MRRs),
 447 allowing the entire multiplication and summation to fit within a single core. After summation, a logarithmic
 448 softmax activation (also assumed to be implemented in analog electronics) classifies the signals to predict the
 449 corresponding digit.

f. May the authors present data showing >8000 cycles endurance?

We have measured the device >8000 times and have attached the raw data in “Source Data.zip” file. The “retention_time_pda255”, “leakage”, “leakage_improved”, “write_time”, and “write_time_pda255” contain raw measurement data writing to the analog memory for each respective experiment (retention time, leakage, and write time).

g. Regarding the crosstalk characterization in Appendix B and Fig. C2, the results indicate that the wavelength shift magnitudes between peak 3 and peaks 1,2, and 4 are comparable but occur in opposite directions. Could authors explain why the peaks shift in opposite directions? Furthermore, it appears that the wavelength shift magnitude is not negligible when only tuning one ring resonator.

Peaks 1,2,4 were biased and held at 0 V while peak 3 was tuned from -1 V to 3 V reverse bias, then several spectra measurements were conducted at each tuned voltage where the resonance wavelengths were extracted. With multiple measurements at the same tuned voltage, the resonance wavelengths were averaged. Therefore, peaks 1, 2, and 4 reveal system level deviation from a nominal resonance wavelength, due to thermal variations, despite using a temperature controller for the chip. Peak 3 separates from the other peaks due to the plasma dispersion effect. We have updated the manuscript to clarify these points:

350 -6 dBm. In Figure C2c and Figure C2d, to observe the tuning efficiency of the PN junction MRRs, one MRR
 351 (peak 3) is biased at various voltages from 0 to 3 V (3 V is the limit for the thick oxide transistors) while the
 352 other MRRs (peaks 1, 2, and 4) are biased at zero voltage. Several spectra measurements are conducted at each
 353 tuned voltage where the resonance wavelengths are extracted. With multiple measurements at the same tuned
 354 voltage, the resonance wavelengths are averaged. Therefore, peaks 1, 2, and 4 reveal system level deviation from
 355 a nominal resonance wavelength, due to thermal variations from measurement to measurement, despite using a
 356 temperature controller for the chip. Peak 3 shifts to longer wavelengths relative to the other peaks because the
 357 increasing reverse bias voltages on this MRR cause a plasma dispersion effect to shift the resonance wavelength.

Reviewer #3 (Remarks to the Author):

Reviewer #4 (Remarks to the Author):

The authors present a new approach to neuromorphic photonic computing, employing analog memory co-located with photonic devices. The paper is technically sound, well organized and well written, but the degree of novelty is moderate in my opinion.

Thank you for your comment. In the revised manuscript, we have tried to highlight our novelty better by addressing your comments below. Furthermore, we have also included EDP calculations of our work and comparison to state-of-the-art neuromorphic photonic accelerators.

Here I have a list of comments that can help the authors improve their manuscript:

- The proposed approach consists of employing a capacitance to store the voltage needed to tune the MRR resonator. In this way, in an “n x n” matrix, instead of employing n² DACs, only n DACs – one for each column – are used, saving power, area and complexity. This approach of employing time-multiplexed DACs followed by hold-capacitors (very popular in standard electronic applications) presents some drawbacks in terms of ratio between retention time and network latency, as the authors highlighted. However, I would appreciate more insights about that, because there are several aspects neglected in this work that can become dominant. The dead-time for refreshing is not only the write time of the single capacitor multiplied the number of capacitors. Also the settling time of the DAC, needed to change its output voltage, could be relevant, as well as the time needed to update the input of the DAC (e.g. the write time of the SRAM preceding the DAC). Please clarify this point.

Thank you for the keen suggestion. We have updated the manuscript to acknowledge that the network latency and dead-time may come from latencies in the DAC settling time, SRAM to DAC delay, and refresh time.

203 Once weights become unusable due to leakage, dead-time is consumed to refresh the weight, which consists
 204 of the time needed to update the DAC inputs, analog memory write time, and DAC settling time. For DEOAM,
 205 the write time is about 65 ns in the fastest case. Surveyed DACs, in similar process nodes to DEOAM, can
 206 achieve sample rates up to 3 GS/s (or 333 ps per sample), and reported SRAM propagation delays are in the
 207 10s of picoseconds, meaning that the time needed to update the DAC inputs and DAC settling time is negligible
 208 compared to the DEOAM write time [18, 19]. Even for the fastest write time among surveyed analog memory
 209 technologies (500 ps for PCMs), the analog memory write time is similar to DAC sample rates [53]. Therefore,
 210 the dead-time can be approximated by the analog memory’s write time. When examining the ratio of retention

- Analogously, in equation (2) - concerning the power consumed by DEOAM - some terms are missing. In the standard approach, the power consumed by each SRAM-DAC is only static, and depending on the kind of DAC, can be very low. With the DEOAM approach, instead, the dynamic power of the DAC can be a relevant portion of the DAC power. Moreover, in DEOAM also the inputs of the DAC are dynamically changed, so also this contribution needs to be taken into account.

Thank you for the feedback. We have updated the expressions for the power consumption to distinguish between static and dynamic power consumption:

$$P_{SRAM-DAC} = n^2(P_{DAC_{static}} + P_{SRAM_{static}}) + n(P_{DAC_{dynamic}} + P_{SRAM_{dynamic}}) \quad (1)$$

$$P_{DEOAM} = n(P_{DAC_{static}} + P_{DAC_{dynamic}} + \alpha C f V^2)$$

Regarding the power consumed by the inputs to the DAC in the DEOAM architecture, if we consider a memory and DAC in both architectures, both architectures consume similar power when modulating the inputs (input to the SRAMs in the SRAM-DAC architecture and input to the DAC in the DEOAM architecture), but the main difference in power consumption is the SRAM to DAC power consumption in the SRAM-DAC architecture, the DAC to DEOAM power consumption in the DEOAM architecture, and the distribution of active DACs between the two architectures. In the SRAM-DAC architecture, power consumption begins when the SRAM changes state and ends when the DAC settles, meaning there is SRAM static and dynamic power and DAC static and dynamic power. In the DEOAM architecture, power consumption begins when the DAC changes state and ends when charge settles on the DEOAM, meaning there is DAC static and dynamic power and DEOAM dynamic power. Finally, there is the distribution of DACs from the architecture that determines the power consumption scaling. We have updated the manuscript to include this description.

93 switching voltage. If we consider a memory element and DAC in both architectures, both architectures consume
 94 similar power when modulating the inputs (input to the SRAMs in the SRAM-DAC architecture and input to
 95 the DAC in the DEOAM architecture), but the main difference in power consumption is the SRAM to DAC
 96 power consumption in the SRAM-DAC architecture, the DAC to DEOAM power consumption in the DEOAM
 97 architecture, and the distribution of active DACs between the two architectures. In the SRAM-DAC architecture,
 98 power consumption begins when the SRAM changes state and ends when the DAC settles, meaning there is
 99 SRAM static and dynamic power and DAC static and dynamic power. In the DEOAM architecture, power
 100 consumption begins when the DAC changes state and ends when charge settles on the DEOAM, meaning there is
 101 DAC static and dynamic power and DEOAM dynamic power. Finally, there is the distribution of DACs from the
 102 architecture that determines the power consumption scaling. DACs are still required in the DEOAM architecture

- The abstract of the manuscript is only qualitative. Some quantitative results would be appreciated.

Thank you. We have added the network latency to retention time ratio of 100 that is necessary to maintain >90% inference accuracy and potential power savings from using DEOAM vs. SRAM-DAC implementation.

Abstract

In neuromorphic photonic systems, device operations are typically governed by analog signals, necessitating digital-to-analog converters (DAC) and analog-to-digital converters (ADC). However, data movement between memory and these converters in conventional von Neumann architectures incur significant energy costs. We propose an analog electronic memory co-located with photonic computing units to eliminate repeated long-distance data movement. Here, we demonstrate a monolithically integrated neuromorphic photonic circuit with on-chip capacitive analog memory and evaluate its performance in machine learning for in situ training and inference using the MNIST dataset. Our analysis shows that integrating analog memory into a neuromorphic photonic architecture can achieve over $26\times$ power savings compared to conventional SRAM-DAC architectures. Furthermore, maintaining a minimum analog memory retention-to-network-latency ratio of 100 maintains >90% inference accuracy, enabling leaky analog memories without substantial performance degradation. This approach reduces reliance on DACs, minimizes data movement, and offers a scalable pathway toward energy-efficient, high-speed neuromorphic photonic computing.

- I recommend including the details of the capacitance implementation, namely the metal-oxide-metal technology and dimensions, within the main body of the work rather than relegating this information solely to the appendix. Also, the

size of the PN-junction MRR should be added in the text, to understand the possibility of scaling up the size of the neural networks.

We have updated the manuscript to describe the capacitance implementation:

119 The neuromorphic photonic hardware with integrated DEOAM is shown in Figure 2. The MRRs are PN junction
120 doped modulators operating in depletion mode (reverse bias) and are connected to analog memory cells consisting
121 of 100x100 μm metal-oxide-metal capacitors that use interdigitated fingers to increase capacitance. The MRRs
122 have different ring perimeters starting with 144.248 μm and 50 μm PN junctions, increasing by 50 nm to avoid
123 resonance collision. Key figures of merit for DEOAM are summarized in Table 1, and measurement results are

- Related to the previous point, it would be helpful to add some considerations about the scalability of the proposed approach with larger networks. Considering the trade-off between refresh time and inference time of the network, what are the reasonable maximum sizes (with the relative area consumption) achievable with this approach?

Thank you for the feedback. We have updated the Discussion section to discuss scalability and reasonable network sizes, which arise from limitations in ring's FSR and finesse, SNR, and refresh time:

231 However, scaling the neuromorphic photonic processor with analog memory presents challenges in analog
232 memory and MRR control. As the network scales, reasonable sizes of the network can be up to 108 rings in a
233 weight bank (limited by the ring's FSR and finesse) and up to a few hundred rows of weight banks (limited by
234 SNR)[49, 50]. The size of the network can also be limited to the analog memory's retention time, refresh time,
235 and inference time (network latency) where a retention time must be allocated to refresh time for all rows of the
236 network and inference time to ensure the network can operate. For DEOAM, the retention time to write time
237 ratio is about 10,000, and inference times (picosecond to nanosecond range) are negligible compared to the write
238 time. Therefore, if SNR is not limiting the number of rows in the network, the next limitation is the refresh rate,
239 which is limited to several thousand rows. Depending on the analog memory requirements, specialized circuitry

- Even if the front-end was not integrated in the prototype, it is important to give more details about that because it also impacts on the overall performance, specially in terms of latency of the network.

The front-end is coming from measurement equipment as shown in Figure A1. We have updated the figure description to clarify the location of the front-end so that readers may look up these equipment specifications for the front-end.

Fig. A1: Measurement setup to characterize DEOAM. a) Retention time measurement setup, b) write time measurement setup, and c) leakage measurement setup. The front-end is in the measurement equipment.

- In the Discussion section, the authors mention “backpropagation hardware”, but there are no details or references about that.

We have removed mentioning backpropagation hardware.